# Bridging Context Gaps: Leveraging Coreference Resolution for Long Contextual Understanding

**Yanming Liu[1], Xinyue Peng[2], Jiannan Cao[4], Shi Bo[3], Yanxin Shen[1], Tianyu Du[1]\*,**
**Sheng Cheng[1], Xun Wang[1], Jianwei Yin[1], Xuhong Zhang[1]\***
[1]Zhejiang University, [2]Southeast University, [3]Boston University
[4]Massachusetts Institute of Technology
{oceann24, ssyysyx, zhangxuhong, zjuyjw, zjradty}@zju.edu.cn
jiannan@mit.edu, xinyuepeng@seu.edu.cn, shibo@bu.edu

## Abstract

Large language models (LLMs) have shown remarkable capabilities in natural language processing; however, they still face difficulties when tasked with understanding lengthy contexts and executing effective question answering. These challenges often arise due to the complexity and ambiguity present in longer texts. To enhance the performance of LLMs in such scenarios, we introduce the Long Question Coreference Adaptation (LQCA) method. This innovative framework focuses on coreference resolution tailored to long contexts, allowing the model to identify and manage references effectively. The LQCA method encompasses four key steps: resolving coreferences within sub-documents, computing the distances between mentions, defining a representative mention for coreference, and answering questions through mention replacement. By processing information systematically, the framework provides easier-to-handle partitions for LLMs, promoting better understanding. Experimental evaluations on a range of LLMs and datasets have yielded positive results, with a notable improvements on OpenAI-o1-mini and GPT-4o models, highlighting the effectiveness of leveraging coreference resolution to bridge context gaps in question answering. Our code is public at https://github.com/OceannTwT/LQCA.

## 1 Introduction

Large language models (LLMs) (Brown, 2020; Chowdhery et al., 2023; Ouyang et al., 2022) have demonstrated exceptional competitiveness across various tasks, including question answering, summarization, and generation (Wang et al., 2023). Recently, many new versions of LLMs have begun to extend their capability to handle longer contexts (Yang et al., 2024; Dubey et al., 2024). Both open-source and proprietary models show some proficiency in understanding long texts and support extended reading (Zhang et al., 2024a). However, LLMs still struggle to accurately identify key passages from the middle of long contexts (Liu et al., 2024) and generate effective responses or expected content based on these passages when overwhelmed by excessive information (Shi et al., 2023). This suggests that despite having larger context windows, many LLMs still face challenges in handling long-text tasks (Li et al., 2024b).

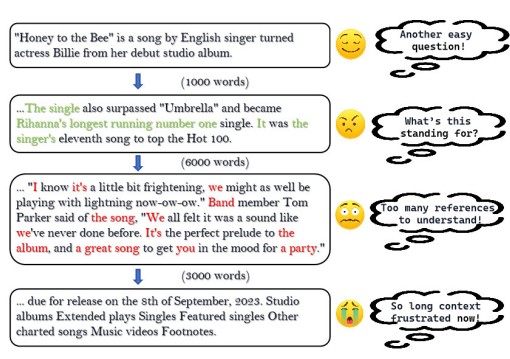

Figure 1: The complexity arises from multiple entities and coreference relations in long contexts, making it difficult for large models to effectively learn contextual information.

---

\*Corresponding author.

Currently, a range of studies are being conducted on long-text reasoning, covering areas such as foundational models for long texts, long-text datasets (Li et al., 2024a; An et al., 2024), task types, and multi-turn dialogues. To more efficiently retrieve contextual information, techniques like retrieval-augmented methods (Phan et al., 2024; Bai et al., 2024; Zhu et al., 2024a) for slicing long contexts or compressing (Huang et al., 2024) and reinterpreting articles are rapidly advancing.

In the domain of information extraction, LLMs still fall short in tasks such as entity extraction, coreference resolution, and event detection, often described as "large but not precise" (Isik et al., 2024; Kumar, 2024). Recent research has attempted coreference resolution through prompt engineering, while some studies based on the DeBERTa (He et al., 2021) model series have shown improvements in coreference resolution tasks (Martinelli et al., 2024). However, these models remain limited by their smaller parameter sizes and constrained context windows (Otmazgin et al., 2023).

Although LLMs exhibit some capabilities in handling long-text tasks, their performance remains highly dependent on the quality of the text. In lengthy texts, the use of numerous similar referring expressions, modifiers, and varying noun phrases can pose significant comprehension challenges for the models. This is especially critical when dealing with ambiguous sentences, where LLMs are prone to confusion (Shi et al., 2023). Therefore, improving the quality of reasoning for long-text tasks and focusing on the quality of long-text content have become critical research topics.

To address the challenges of LLMs in understanding long contexts and question answering, we propose the **L**ong **Q**uestion **C**oreference **A**daptation (**LQCA**) method, an efficient framework for long-context coreference resolution and question answering. The LQCA framework is capable of resolving all mentions in long contexts and selecting the best answer mention to replace the original text. This framework consists of four steps: coreference resolution on sub-document, mentions distances computation, defining coreference representative mention, and question answering with mention replacement. These four steps systematically process information within long contexts, ultimately replacing it with partitions that are easier for large models to understand. We conducted experiments on five large models with varying parameter sizes and nine datasets focused on long-context questions. The results indicate that long contexts processed through coreference resolution perform best during inference with language models, achieving a 3.61% improvement on GPT-4. Our method demonstrates the advantages and capabilities of information extraction and coreference resolution in understanding long contexts while emphasizing the importance of text quality for model inference.

**Our Contributions.** Our main contributions can be summarized as follows:

- We propose the LQCA framework, a coreference resolution method designed to enhance the LLM capability of long context responses. Our approach partition the context and merges coreference information to achieve comprehensive coreference resolution results for the entire document, replacing segments of the text to improve its quality, thereby helping the model better understand the context.

- We demonstrate that text quality plays a crucial role in answering questions related to long text comprehension. By optimizing text quality and eliminating ambiguity, we enable the model to learn and reason more effectively for the target tasks.

- Extensive experiments show that our method performs exceptionally well across various tasks, including long text question answering and long text classification, providing a new paradigm for addressing long text issues.

## 2 PRELIMINARIES

**Coreference Resolution and Mentions.** In information extraction and dialogue systems, traditional methods involve dealing with entities, relationships between entities, and semantic slot filling when processing context. However, in certain contexts and dialogue scenarios, such as speaker recognition and long context reading, the context often contains a large number of pronouns and modifiers, which significantly affects the model's understanding of the original information. To address this situation, the context can be optimized through two steps. The first step is mentions extraction: Given the input text $x$, extract all pronouns, nouns, noun phrases, and modifiers in the context to obtain the mention set $\mathcal{M} = \{m_1, m_2, \ldots, m_k\}$. For a specific mention $m_i$, its reference information is represented as $R(m_i) \to m_i$. In the input text, the reference information is likely to have a large number of

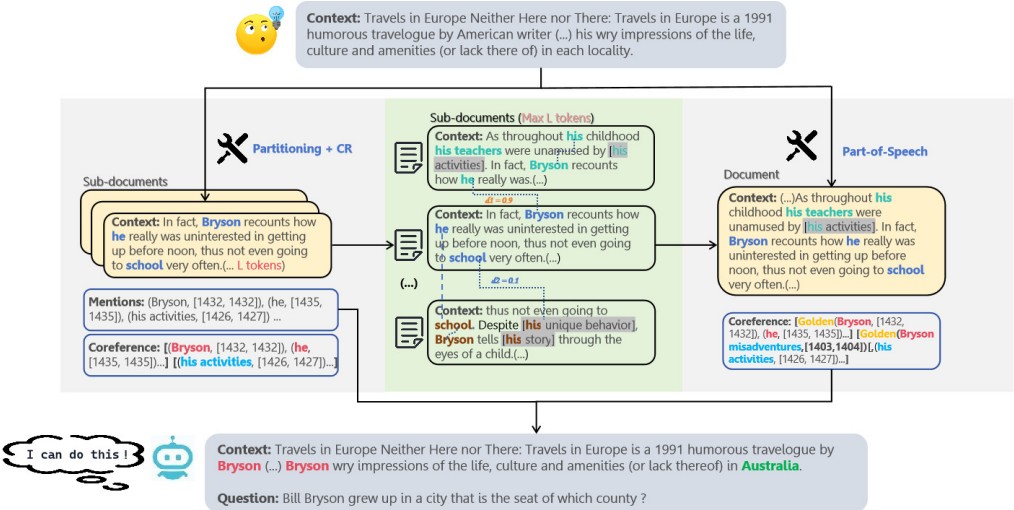

Figure 2: Our Long Question Coreference Adaptation(LQCA) method addresses the coreference resolution problem in long texts through four steps, leveraging indirect coreference and mention relations, and providing the resolved context to the LLM for question reasoning.

meaning-redundant references within the mention set, referred to as co-reference $c$. For two meaning-redundant mentions $m_i$ and $m_j$, it holds that $\{R(m_i), R(m_j)\} \subseteq c$. Since mentions inherently refer to themselves, this can also be expressed as $\{m_i, m_j\} \subseteq c$. By obtaining all coreference relationships and replacing mentions in the original text, the clarity of the text can be effectively enhanced.

## 3 METHODOLOGY

We employ a four-step Long Question Coreference Adaptation (LQCA) method. Each step involves the integration and extraction of information from the text. By partitioning long documents and resolving references within each sub-document, we use clustering techniques to merge the results from different sub-documents into the same cluster, thereby achieving overall reference resolution for the entire document. Finally, we replace the references and provide the cleaned text to the large model for contextual understanding and response.

### 3.1 COREFERENCE RESOLUTION ON SUB-DOCUMENT

The state-of-the-art reference resolution model, Maverick (Martinelli et al., 2024) is based on the DeBERTa-v3 (He et al., 2021) architecture, which has specific requirements regarding the length of the input tokens.We cannot directly perform reference resolution on the entire long input. Given an long input context $X$, we initiate the process by partitioning the context into sub-documents $\{S_i\}_{i=1}^N$, each constrained to a maximum length of $L$. $N$ is the total number of partitioning documents. Our partitioning is based on a sliding window approach, where each partitioning starts from the beginning of a sentence and extends to the position of the last sentence that does not exceed the length of $L$. If the last sentence exceeds this limit, truncation is applied for the whole sentence.

For each sub-document $S_i$, we utilize the maverick-mes-ontonotes[1] model to perform mentions extraction and coreference resolution. $\mathcal{M} = \{m_1, m_2, \ldots, m_k\}$ represent the set of detected mentions in the input sequence, where each mention $m_i$ is characterized by its position $p_i$. In the coreference resolution of sub-documents, $c_i^j$ represents the $j$-th coreference set in the $i$-th sub-document and $C_i = \{c_i^1, c_i^2, \ldots, c_i^j\}$ is the total coreferences in single sub-document. The mentions under this coreference set belong to this reference. After providing the sub-documents to the Maverick model through batch inference, we obtained all the coreferences $\mathcal{C} = \{C_i\}_{i=1}^N$ and mention $\mathcal{M}$ corresponding to the long text. The results help us in the subsequent processing of recognizing and merging the same coreferences and mentions, allowing us to combine the information from these sub-documents.

---

[1]https://huggingface.co/sapienzanlp/maverick-mes-ontonotes

## 3.2 Mentions Distances Computation

To address the issue of co-referent mentions being distributed across multiple partitions, we construct a sparse matrix representing mention relationships. For any two mentions $m_a$ and $m_b$ occurring in the same sub-document, we define their co-reference score $s_i$ under a particular sub-document and a specific reference $j$ as follows:

$$s_i^j(m_a, m_b) = \begin{cases} 1 & \text{if } m_a, m_b \subseteq c_i^j \\ 0 & \text{otherwise} \end{cases}$$

Similarly, the score for non-coreference $t_i$ is defined as:

$$t_i^j(m_a, m_b) = \begin{cases} 1 & \text{if } m_a, m_b \nsubseteq c_i^j \\ 0 & \text{otherwise} \end{cases}$$

Thus, the co-reference score for these two mentions within the sub-document is $s_i(m_a, m_b) = \sum_j s_i^j$, and the non-co-reference score is $t_i(m_a, m_b) = \sum_j t_i^j$. Since two mentions can only belong to the same reference in a single sub-document, the scores $s_i(m_a, m_b)$ and $t_i(m_a, m_b)$ satisfy $s_i(m_a, m_b) \oplus t_i(m_a, m_b) = 1$, indicating whether the two mentions are in the same reference.

Through this process, we can determine the co-reference relationship of any two mentions within a sub-document. Only when both mentions are present in the sub-document can they receive a mention score. For cross-sub-document mention relationships, we integrate information across multiple sub-documents based on the distance between the two mentions.

For the entire long context, we define $d(m_a, m_b)$ as the distance between two mentions. Since mentions that appear in the same sub-document have already had their co-reference relationship evaluated by the Maverick model, we can assess their distance to determine whether they refer to the same entity in the context of the long document. The score between mentions is computed as:

$$d(m_a, m_b) = \frac{\sum_{i=1}^{N} s_i(m_a, m_b)}{\sum_{i=1}^{N} s_i(m_a, m_b) + \sum_{i=1}^{N} t_i(m_a, m_b)}$$

For mentions not in the same sub-document, since they lack coreference and coreference scores, merging their relationships requires evaluation based on distance information. For any mention $m_c$, if the distance between mentions $m_a$ and $m_c$ is $d(m_a, m_c)$, and the distance between mentions $m_b$ and $m_c$ is $d(m_b, m_c)$, we apply the multiplication principle:

$$d(m_a, m_b) = \max_{c \in M}\{d(m_a, m_c) \times d(m_b, m_c)\}$$

Using this distance evaluation, the distance information between any two mentions in the mention cluster can be calculated. This form can be easily computed using the variations of the shortest path problem to obtain the corresponding distance information. The algorithm could be refered in Appendix G.

After computing the longest dot product path, we compare the longest path score to a predetermined threshold $k$, and based on this information, we construct a mention relationship graph $G(\mathcal{M}, E)$:

$$(m_a, m_b) \in E, \text{if } d(m_a, m_b) > k,$$

In the mention relationship graph, each strongly connected component $C_i = \{u \in M : \forall v \in C_i, u \leftrightarrow v\}$ represents a set of coreferent mentions.

This classification helps bridge the gaps in context, ensuring that relevant mentions are identified as referring to the same entity, thereby improving the model's understanding of the document content.

### 3.3 DEFINING COREFERENCE REPRESENTIVE MENTIONS

For the strongly connected components we obtained, each strong connection serves as a coreference, and when inputting into the LLM later, all mentions under the same coreference need to be replaced, especially those mentions that are pronouns, which should be transformed into specific, meaningful content.

We first use the lightweight spaCy model en_core_web_sm to perform part-of-speech tagging on all words in the article. In the tagging results, if a token in the mention span belongs to PRON, it is marked as $p(m_i) = \text{PRON}$. Additionally, for a coreference, we want the transformed text to select the mention that contains the most meaningful equivalent. Let $m_{c_i}$ be the representative mention of reference $C_i$, and $f(m_k, c_i)$ be the number of times mention $m_k$ appears under coreference $c_i$. The process could formulated as:

$$m_{c_i} = \text{argmax}_{m_k}\{f(m_k, c_i) \times [p(m_i) \neq \text{PRON}]\}$$

If there are multiple mentions that satisfy this condition, we select the earliest mention based on the principle of first selection. This representative mention serves as the normalized text for the entire coreference. By effectively replacing vague mentions with their n the text, enabling LLMs to maintain a coherent understanding of the context.

### 3.4 QUESTION ANSWERING WITH MENTIONS REPLACEMENT

We need to modify the original text into the target text by replacing all mentions in the text based on each coreferent representative mention. When handling overlapping cases, we prioritize replacing the mention with the largest overlapping range. If no suitable replacement is available (for example, only pronouns), we retain the original text.

To execute the Question Answering (QA) task, we leverage the reasoning capabilities of the LLM, formalized in the following process:

$$\hat{R} = \arg\min_R \left( -\log P(R|C', Q) \right)$$

where $P(R|C', Q)$ represents the probability of generating a response $R$ given the modified context $C'$ and the question $Q$. By implementing these steps, our approach effectively addresses the coreference issues in long texts, enabling subsequent tasks (such as QA) to interact more accurately with the content and enhance contextual awareness. This integrated method not only tackles the challenges posed by long texts but also effectively bridges contextual gaps, improving LLM performance in understanding long contexts.

## 4 EXPERIMENTAL SETUP

### 4.1 DATASETS

To evaluate the performance of LQCA in long text contexts, we primarily conduct assessments on question-answering data across three datasets, targeting different task categories including summarization tasks, question-answering tasks, and multiple-choice classification tasks. Specifically, these three datasets are LooGLE (Li et al., 2024a), L-Eval (An et al., 2024), and LongBench (Bai et al., 2024). For the summarization task, we select the arXiv paper abstract category from the LooGLE dataset for evaluation. For the question-answering task, we focus on closed-ended tasks in L-Eval, emphasizing multiple-choice and true/false question-answering tasks. For the dataset requiring information extraction from texts, we utilize the multi-document question-answering dataset under LongBench, primarily selecting three English-based datasets: HotpotQA (Yang et al., 2018), 2WikiMultihopQA (Ho et al., 2020), and MuSiQue (Trivedi et al., 2022). This diverse selection enables us to better understand the performance of our approach across different tasks and datasets, providing a comprehensive perspective on its strengths and weaknesses in multi-task scenarios. For more details on the datasets, please refer to Appendix B.

## 4.2 BASELINES

Currently, methods for addressing long-context issues mainly involve guiding models through chain-of-thought techniques for specific step reasoning. One method is the compression strategy, which condenses key information from long texts into concise paragraphs, allowing questions to be answered and processed based on this. Another commonly used method involves slicing text segments for retrieval. Our comparison benchmarks include the following methods:

**Vanilla LM**: Guides the model to generate responses through simple text input, testing its basic ability to handle long contexts.

**Zero-shot Chain-of-Thought** (Kojima et al., 2022): Utilizes prompt templates, designing specific prompts to guide the model's reasoning, leveraging the model's inference capabilities without the need for additional training data.

**RAG** (Lewis et al., 2020; Ram et al., 2023) **with Self-Corpus**: Unlike traditional RAG, we do not introduce additional knowledge but instead use slices of long text segments as the corpus. When questioning relevant issues, we retrieve from the corpus and combine it with context for reasoning, aiming to improve the model's understanding and response quality to long texts.

**RECOMP** (Xu et al., 2024) **with Self-Corpus**: Builds upon RAG with an added compression strategy, using the same long text segment slice corpus. This method improves processing efficiency by compressing key information from long texts, aiming to optimize the model's reasoning process without losing important information.

## 4.3 MODELS

We utilize five models, including three from OpenAI's GPT series (Brown, 2020): `o1-mini-2024-09-12`, `gpt-4o-2024-08-06`, and `gpt-4o-mini-2024-07-18`, as well as two open-source large models from Llama-3 (Dubey et al., 2024) and Qwen-2 (Yang et al., 2024): `llama3-gradient-8b` and `qwen2-7b`. The five tested large models all have a 128k context window, which meets our requirements for evaluating and testing long context datasets, reducing the information loss caused by additional text segmentation. Comparing different models helps us understand whether their core reference performance in long contexts is similar.

## 4.4 IMPLEMENTATION

Our experiments were primarily evaluated in a zero-shot setting. For inference predictions of general models, we used default settings and adopted greedy search as the inference benchmark. For zero-shot reasoning chains, we referred to the prompt templates as the default chain-of-thought scheme. The evaluated models have a long context window, allowing us to retain almost all contextual information for the questions without truncation.

Nonetheless, we still establish a safety mechanism for ultra-long texts. When the input length $L$ exceeds the model's maximum context length (indicated by the name suffix), we truncate from the middle of the input sequence $S$ to avoid losing the beginning and end portions, which may contain key information. During the generation process, we employed greedy decoding to ensure the reproducibility of results. For LQCA, our default experimental setting is a coreference score threshold of 0.9, meaning if the distance between two mentions exceeds 0.9, we consider them within the same coreference. For the spaCy model, we used en_core_web_sm model for tokenization and part-of-speech tagging.

In terms of benchmark comparison, we use a retrieval method, with *Contriever-msmarco* (Izacard et al., 2022) as the retrieval model responsible for slicing articles in a sliding, non-overlapping manner, with each segment limited to 512 tokens. Additionally, in the RECOMP experiments, we choose the abstract compression tool provided by RECOMP as the compression tool to help the model effectively extract important information from text segments. The comparison of LQCA with other baselines is based on the differences in the comparison datasets, evaluated using Rouge-L, accuracy and F1 scores.

Table 1. Evaluations of the performance of different models and methods across three datasets. Assess the long context ability in question answering, summarization, and knowledge extraction.

| Models | LooGLE | | L-Eval | | | | LongBench | | | Avg. |
|---|---|---|---|---|---|---|---|---|---|---|
| | Arxiv paper abstract (Rouge-L) | Long dependency QA (Rouge-L) | TOEFL (Acc.) | QuALITY (Acc.) | Coursera (Acc.) | SFcition (Acc.) | HotpotQA (F1) | 2WikiMultihopQA (F1) | MuSiQue (F1) | |
| *Vanilla LM* | | | | | | | | | | |
| OpenAI-o1-mini | 35.62 | 33.98 | 90.55 | 89.14 | 85.44 | **88.94** | 68.33 | 49.53 | **39.83** | 64.60 |
| GPT-4o | 30.47 | 27.46 | 88.64 | 86.49 | 80.12 | 80.55 | 64.12 | 47.99 | 35.55 | 60.15 |
| GPT-4o-mini | 27.95 | 25.61 | 85.61 | 87.11 | 75.98 | 80.26 | 62.57 | 40.71 | 33.97 | 57.75 |
| Llama-3-8b | 6.25 | 9.13 | 66.19 | 52.69 | 46.22 | 60.94 | 44.56 | 32.55 | 23.51 | 38.00 |
| Qwen2-7b | 7.56 | **6.64** | 57.35 | 48.92 | 50.79 | 62.33 | 46.19 | 34.11 | 20.69 | 37.17 |
| *Zero-shot CoT* | | | | | | | | | | |
| OpenAI-o1-mini | 34.89 | 32.56 | 89.42 | 88.55 | 85.97 | 87.66 | 71.55 | 50.24 | 37.85 | 64.30 |
| GPT-4o | 31.25 | 28.75 | 88.91 | 87.12 | 82.15 | 81.46 | 66.94 | 49.36 | 34.95 | 61.21 |
| GPT-4o-mini | 28.56 | 26.11 | 86.14 | 86.23 | 76.33 | 81.03 | 63.23 | 41.97 | 34.12 | 58.19 |
| Llama-3-8b | 5.66 | **11.23** | 62.59 | 55.39 | 46.49 | 62.09 | 42.90 | **34.92** | 24.90 | 38.46 |
| Qwen2-7b | 6.96 | 7.12 | 58.73 | 47.66 | 49.23 | 61.74 | 43.56 | 35.95 | **22.61** | 37.06 |
| *RAG with self-Corpus* | | | | | | | | | | |
| OpenAI-o1-mini | 30.95 | 30.12 | 86.55 | 85.03 | 83.64 | 84.98 | 70.26 | 48.23 | 36.73 | 61.83 |
| GPT-4o | 27.65 | 22.35 | 85.13 | 80.12 | 80.65 | 78.12 | 66.35 | 47.01 | 32.11 | 57.72 |
| GPT-4o-mini | 25.71 | 23.14 | 83.22 | 82.23 | 73.92 | 79.19 | 62.94 | 42.55 | 33.52 | 56.26 |
| Llama-3-8b | 6.84 | 8.56 | 63.15 | 45.66 | 48.77 | 52.94 | 40.72 | 31.78 | 22.03 | 35.60 |
| Qwen2-7b | **8.33** | 5.23 | 58.79 | 43.12 | 46.21 | 58.39 | 41.80 | 34.69 | 19.42 | 35.10 |
| *RECOMP with self-Corpus* | | | | | | | | | | |
| OpenAI-o1-mini | 29.56 | 28.11 | 85.96 | 83.91 | 79.46 | 85.30 | 68.91 | 45.37 | 29.14 | 59.52 |
| GPT-4o | 28.13 | 21.55 | 85.13 | 78.52 | 76.25 | 77.66 | 65.02 | 44.03 | 30.32 | 56.28 |
| GPT-4o-mini | 24.22 | 20.96 | 82.49 | 79.11 | 74.75 | 75.97 | 63.55 | 42.59 | 27.19 | 54.53 |
| Llama-3-8b | **7.11** | 10.20 | 55.43 | 46.17 | 42.30 | 54.65 | 41.95 | 30.58 | 20.63 | 34.33 |
| Qwen2-7b | 7.26 | 5.23 | 56.17 | 44.96 | 45.19 | 57.92 | 41.62 | 32.91 | 17.95 | 34.35 |
| *Long Question Coreference Adaptation(LQCA)* | | | | | | | | | | |
| OpenAI-o1-mini | **40.65** | **37.26** | **93.71** | **91.66** | **87.25** | 88.24 | **75.92** | **56.85** | 38.55 | **67.78** |
| GPT-4o | **34.53** | **30.17** | **91.03** | **89.62** | **85.12** | **81.92** | **71.66** | **53.62** | **36.25** | **63.76** |
| GPT-4o-mini | 29.59 | **28.45** | **88.72** | **88.65** | **83.99** | 79.88 | **70.33** | 46.16 | **34.82** | **61.17** |
| Llama-3-8b | 6.91 | 9.22 | **69.55** | 54.67 | **49.15** | 58.94 | **46.23** | 34.72 | **25.03** | **39.37** |
| Qwen2-7b | 7.51 | 5.79 | **61.49** | 50.22 | **53.76** | 61.02 | **47.11** | **37.16** | 21.98 | **38.44** |

# 5 EXPERIMENTS

## 5.1 MAIN RESULTS

We evaluate the performances of five LLMs on multiple datasets with LQCA and baselines, and the results are shown in the Table 1.

**Context with coreference resolution.** LQCA has demonstrated highly competitive performance among other long context processing methods. It achieves the best results on almost all datasets when applied to LLMs with a higher number of parameters and strong contextual understanding, consistently outperforming other baseline performances on GPT-4o. Meanwhile, the other two models also achieve the best performance on 7-8 out of 9 baselines methods, respectively. Compared to directly providing the question to the LLM, LQCA achieves an average improvement of **+3.61%** on GPT-4o. On the latest o1-mini model, the coreference resolution method also significantly improves by an average of **+3.18%** across various metrics compared to other methods. At the same time, LQCA has also achieved good performance on open-source large models with relatively small parameter counts, showing improvements in metrics compared to direct prompting and zero-shot chain-of-thought methods. However, due to differences in context understanding capabilities, retrieval methods can sometimes find corresponding answers more accurately for these types of large models. Overall, texts that have undergone coreference resolution can help models understand contextual dependencies to varying degrees, thereby assisting models in solving various problems.

**LQCA in Long dependency question.** When addressing tasks involving long-dependency issues, such as summarization,long dependency QA, and strategy-related question-answering (HotpotQA, 2WikiMHQA) tasks, coreference resolution methods demonstrate a clear advantage over other approaches, especially when the model possesses stronger contextual understanding capabilities. For example, in summarization tasks, the O1-mini model improved performance by **+5.05%** compared to other methods. Even more notably, in datasets like HotpotQA, which contain extensive text passages and dependencies, the impact of coreference resolution is even more pronounced. For instance, on this dataset, GPT4o showed a **+7.59%** improvement in results after incorporating coreference resolution, outperforming other methods by about **+6.5%**. The most challenging aspect of long-dependency tasks is determining various coreference relations and the specific entities being referred to. When a question is directly presented to the model, it may confuse these coreferences, leading to incorrect

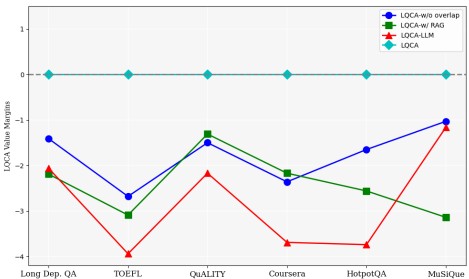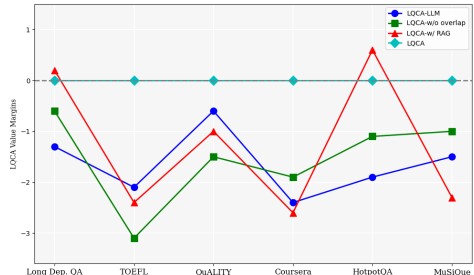

Figure 3: The difference between various coreference resolution variants and the LQCA method on GPT-4o(left) and Llama-3-8b(right).

answers. Thus, effective coreference resolution enhances contextual coherence, helping the model provide more accurate responses.

**LQCA in Knowledge-based question.** For knowledge-based question answering tasks, the improvement brought by coreference resolution methods is not as significant compared to handling long-dependency issues. However, when dealing with long-text knowledge-based question answering, clarifying the references within the text helps enhance the generalization ability of the model. In addressing scientific questions, certain inherent knowledge may have specific representations and specialized terminology, where the model's comprehension ability becomes more important than coreference resolution. Additionally, effective information compression and retrieval strategies contribute to providing accurate answers in commonsense question answering. In commonsense QA, based on four L-Eval datasets, the Llama3-8b model demonstrates an improvement of **+8.44%** compared with RECOMP method, indicating its ability to enhance the contextual understanding of smaller models.

## 5.2   IMPACT OF DIFFERENT COREFERENCE REALIZATIONS

To compare the performance of different coreference resolution designs and the LQCA framework in long document comprehension, as well as to explore the specific roles of each component, We adopt three variations of LQCA as follows:

- *LQCA-LLM*: We use a LLM to perform coreference resolution on each document slice, while other steps remain unchanged. The prompts used for the LLM are provided in Appendix C.

- *LQCA-w/o overlap*: This slicing method segments the document into non-overlapping slices, each no longer than 512 tokens. The coreference resolution results for each slice are used to replace information in the original text, following the same replacement methods as in Defining Representation and Mention Replacement.

- *LQCA-w/ RAG*: After replacing the text, we introduce a *Contriever-msmarco* retriever during the question-answering phase. The retriever uses the corpus provided by the document slices, similar to the setup of the baseline method. Compared to the original LQCA method, this approach reduces the model's dependency on extended context length.

As shown in Figure 3, we evaluate different methods. The coreference resolution method using the LLM shows significant shortcomings in answering questions, likely because the model is currently unable to effectively handle specific downstream tasks. In complex contextual environments with extensive coreference information, expert-level models are often required to handle and annotate mentions accurately. The non-overlapping slicing replacement method performs similarly to LQCA in most tests, but the inability to merge coreference information between different sub-documents slightly limits the question-answering effectiveness for long documents.

Additionally, the *LQCA-w/ RAG* method helps smaller models extract relevant information and even outperforms the LQCA framework on certain metrics. This suggests that retrieval augmentation is more suitable for models with limited capabilities and computational resources in long-document QA tasks. By gathering more relevant information related to the question, it helps the model provide

better answers. The experimental results of these variants further validate the effectiveness of LQCA framework.

## 5.3 SETTING IN SUB-DOCUMENTS

In LQCA, two critical parameters have a significant impact on the coreference resolution performance of the framework. The first is the length of the text segmentation. We ensure that the segmented text length stays within the token limit of the model's input. The second is the preset threshold parameter K, which largely determines whether two mentions refer to the same entity. Due to variations between different texts, the number of sub-documents generated by segmentation may differ, which influences the model's inference performance. To address this, we conduct a grid search on these two variables to analyze their impact on performance, with the experimental results shown in the Figure 4.

**Longer sub-documents help capture contextual information**, improving coreference resolution performance. For example, in the figure, when the sub-document length approaches the upper limit of 512, the model shows good performance with most F1 scores above 70 when the k-value is less than 0.8. Since we resolve coreferences across multiple documents and establish connections between mentions, longer contexts provide more precise information.

**Coreference scores threhold is length affected.** The impact of the score threshold between mentions across the entire document on the framework's performance is strongly correlated with the subdocument length. When the segmented sub-documents are shorter, a lower $k$ value helps maintain consistency between related mentions. However, if the threshold is set too high, some pronouns may not find the correct referent. On the other hand, an excessively high threshold may degrade coreference resolution performance, as incomplete information in certain sub-documents may cause the model to misclassify mentions or assign them to separate categories. Therefore, a moderate $k$ value helps the framework achieve optimal performance.

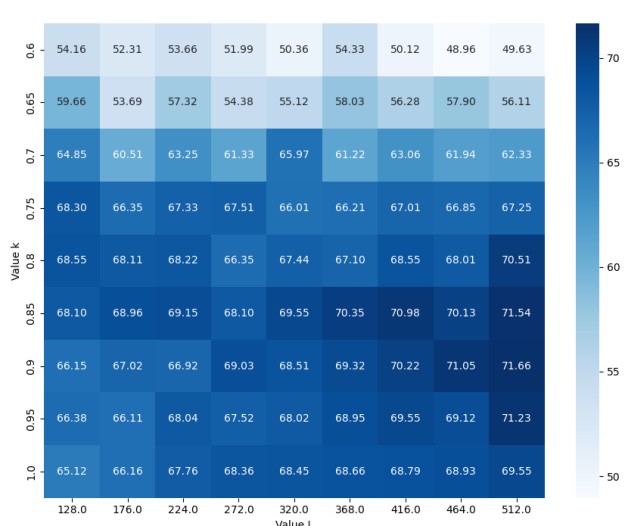

Figure 4: Performance of the LQCA method on the LongBench-HotpotQA dataset using the GPT-4O model under different values of $k$ and $L$. We use the F1 score as the evaluation metric.

## 5.4 EVALUATION OF KEY INFORMATION POSITIONS

Since current large language models tend to focus more on the beginning and the end of a document in long-text scenarios, they often overlook the middle sections where answers may be located. To better assess the effectiveness of our framework, we conducted inference evaluations using the GPT-4o model on the Coursera dataset, marking the percentage position of answers within the text. We divided the positions of the answers into five intervals, using the percentage value at the end of each interval as a marker.

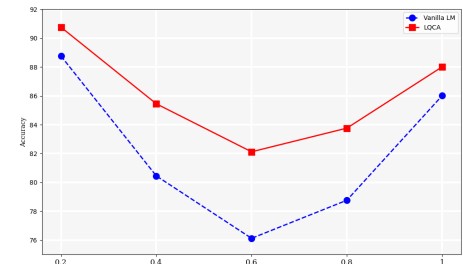

Figure 5: The effect of the position of relevant information on Coursera QA dataset performance.

**Middle position performace is highly improved by LQCA**. As shown in the Figure 5, the model with coreference resolution significantly alleviates the issue of ignoring mid-document

information. It can be observed that after applying coreference resolution, the accuracy of key information in the [0.2, 0.4], [0.4, 0.6] and [0.6, 0.8] intervals improve significantly. Specifically, the accuracy of key information in the [0.4, 0.6] interval increased by **+6.1%** compared to directly providing the context. This indicates that providing accurate coreference relations enhances the quality of context, reducing information loss and contextual ambiguity, thereby enabling the language model to perform more effectively and accurately when answering questions.

# 6 RELATED WORK

## 6.1 LONG CONTEXT UNDERSTANDING IN LLM

Recent advancements in LLMs highlight the significance of long-context understanding across diverse applications (Xiong et al., 2024; Pan et al., 2024). Zhu et al. (2024a) explores strategies to extend the context window of embedding models to 32k tokens without additional training, enabling their application in tasks with long inputs such as legal contracts. LongBench, L-eval and LooGLE serve as multitask benchmarks, providing a comprehensive evaluation of long-context understanding (Bai et al., 2024; An et al., 2024; Li et al., 2024a). a new context understanding benchmark has been proposed, validating the need for improved generative models in understanding context amidst varying training conditions (Zhu et al., 2024b). The study of temporal complex events through LLMs sheds light on the ability to analyze event chains effectively, particularly when utilizing suitable retrieval mechanisms (Zhang et al., 2024b). However, challenges persist; current LLMs struggle with contextual relevance based on information positioning, as indicated by performance degradation in identifying important details (Liu et al., 2024). Our LQCA method improves text quality through coreference resolution, thereby enhancing the long context understanding of tasks by LLMs.

## 6.2 COREFERENCE RESOLUTION

Coreference resolution is an important task in the field of information extraction (Lee et al., 2017; Dobrovolskii, 2021). Techniques such as finetuning pretrained seq2seq transformers have proven effective, where document inputs are mapped to coreference-tagged sequences, emphasizing the importance of model size and supervision levels (Zhang et al., 2023). Furthermore, a novel approach focusing on event coreference emphasizes learning from events rather than entities, integrating multiple representations for improved resolution (Yao et al., 2023a;b). The complexity of coreference evaluation is addressed by highlighting the need for standardized measurement methodologies across different datasets (Porada et al., 2023). Meanwhile, prompt-based methods like CorefPrompt allow for the modeling of events and coreference simultaneously through a masked language model setup (Xu et al., 2023). Linguistic insights contribute to performance enhancements by categorizing mention-pairs into distinct decision types (Otmazgin et al., 2023). We combine coreference resolution with long context, improving text quality by replacing referents in long context texts, which aids in enhancing downstream tasks.

# 7 CONCLUSIONS

This paper presents the LQCA method, a framework aims at enhancing long-context understanding in LLMs by leveraging coreference resolution. The framework operates through four systematic steps: resolving coreferences within sub-documents, calculating mention distances, defining a representative mention for coreferences, and performing question answering with mention replacement. By processing long contexts in this manner, the method simplifies the information, making it more comprehensible for the language models. Experiments conducted on five large models and nine long-context question datasets show a notable improvement during inference, with a recorded 3.61% enhancement on GPT-4o. The findings illustrate the effectiveness of integrating coreference resolution with information extraction to improve comprehension of lengthy texts. Text quality remains a critical factor influencing model inference performance. We believe this framework will contribute to the long-term development of the long context understanding of LLMs.

## 8 ACKNOWLEDGEMENTS

This work was partly supported by the NSFC under No. 62402418 and No. 62102360. This work was also partly supported by the Key R&D Program of Ningbo under No. 2024Z115.

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

## A BROADER IMPACT AND LIMITATIONS

**Broader Impact.** Using coreference resolution methods to optimize long contexts inference is another significant improvement in information extraction within dialogue systems powered by large language models. We believe that by introducing more techniques, such as semantic recognition and entity extraction, large models can achieve better performance when handling long texts. Additionally, we can further investigate the integration of current compression and retrieval-augmented methods with coreference resolution techniques, exploring more potential solutions to enhance text quality. Higher-quality question texts help us better address various downstream tasks and provide appropriate solutions.

**Limitation.** LQCA presents innovative advancements in addressing long-context understanding and question answering. However, certain limitations warrant consideration. Firstly, the coreference resolution might struggle in dealing with ambiguous references or contexts with high complexity, possibly leading to inaccurate mention replacements. Secondly, while the method improves performance on long-context questions, its effectiveness may diminish when applied to shorter contexts, where the overhead of processing could outweigh the benefits. Furthermore, the reliance on pre-defined mention distances may limit adaptability to varied linguistic structures and usages across different domains. Future endeavors may focus on enhancing coreference resolution techniques and exploring adaptive approaches to better manage diverse contexts and improve robustness in various settings.

# B DATASET DETAILS

## B.1 LOOGLE

LooGLE (Li et al., 2024a) is a benchmark specifically designed to evaluate large language models (LLMs) in long-context understanding tasks. This benchmark emphasizes tasks that rely on both short-term and long-term dependencies in text inputs, such as question answering (QA), summarization, and cloze tasks. LooGLE supports automated evaluation metrics, such as BLEU, ROUGE, METEOR, and BERTScore, to assess model performance. Notable baseline models include GPT-4-32K, GPT-3.5-16K, and ChatGLM2-6B-32K, which have been evaluated on short-term dependency tasks like cloze tests and long-term dependency tasks like document summarization.

The benchmark provides detailed configurations on how to optimize large language models for long-context tasks, with a particular focus on retrieval-based tasks and long-form generation tasks.

## B.2 L-EVAL

L-Eval An et al. (2024) is part of a growing suite of long-text understanding evaluation tools. It is designed for multilingual evaluation, testing the performance of large language models on diverse tasks across language families. These tasks include document-level question answering, summarization, and cloze tasks, with a particular focus on the models' ability to handle large amounts of contextual information. The datasets are derived from real-world domains, such as scientific papers, narratives, and technical reports, making L-Eval highly valuable for tasks that require extensive context retention. L-Eval is an ideal testing platform for evaluating models in scenarios like multi-document retrieval, cross-document summarization, and multi-hop question answering, where retaining long and diverse information is crucial.

## B.3 LONGBENCH

LongBench (Bai et al., 2024) is a bilingual, multi-task benchmark designed to test large language models' ability to handle long contexts, covering both English and Chinese. It evaluates model performance on tasks such as narrative understanding, multi-domain question answering, and summarization, where the datasets require handling large amounts of complex input, such as legal documents, scientific reports, and news articles.

This benchmark includes datasets like NarrativeQA (Kočiskỳ et al., 2018), Qasper (Dasigi et al., 2021), HotpotQA (Yang et al., 2018), and DuReader(He et al., 2018), offering diverse application scenarios from multi-document retrieval to entity tracking in long narratives. The models are evaluated not only on single-document tasks but also on multi-document tasks and zero-shot performance tests. LongBench also incorporates Chinese-specific datasets, further extending its applicability in multilingual scenarios.

# C PROMPT FOR VARIATION OF LCQA

---

**LCQA-LLM**

Please analyze the following context, do coreference resoulution. identify the mentions, and replace them with their corresponding golden mentions which have their actual reference and meaning. Mentions could be Pronouns, Nouns, Noun Phrases or Modifiers. Ensure that the revised text maintains the original meaning and reads naturally. Please only output result only contains revised context, don't output any other information. Here is the text to be processed:
**Context**: [Sentence $S$]
**Result**:

---

Table 2. Experiments on LQCA, RAG, and Variation in perturbations of coreference in LQCA.

| Model | Methods | LooGLE-arxiv-paper-abstract | L-Eval-TOEFL | L-Eval-Coursera | LongBench-HotpotQA | LongBench-2WikiMHQA |
|---|---|---|---|---|---|---|
| OpenAI-o1-mini | RAG with self-Corpus | 30.95 | 86.55 | 83.64 | 70.26 | 48.23 |
| | LQCA-with 5% mentions replacement | 32.51 | 88.22 | 82.98 | 71.66 | 47.55 |
| | LQCA | 40.65 | 93.71 | 87.25 | 75.92 | 56.85 |
| GPT-4o | RAG with self-Corpus | 27.65 | 85.13 | 80.65 | 66.35 | 47.01 |
| | LQCA-with 5% mentions replacement | 28.54 | 82.51 | 81.35 | 68.77 | 47.23 |
| | LQCA | 34.53 | 91.03 | 85.12 | 71.66 | 53.62 |
| Llama-3-8b | RAG with self-Corpus | 6.84 | 63.15 | 48.77 | 40.72 | 31.78 |
| | LQCA-with 5% mentions replacement | 6.56 | 64.25 | 47.29 | 41.29 | 33.69 |
| | LQCA | 6.91 | 69.55 | 49.15 | 46.23 | 34.72 |

## D    PERTURBATIONS ON COREFERENCE

To assess the effectiveness of our algorithm on the accuracy of coreference, our method demonstrates an effective filtering mechanism specifically designed to handle highly ambiguous references. Specifically, in our algorithm, when the similarity between two mentions exceeds a set threshold, we ensure that at least 9 out of the 10 subdocuments containing both mentions classify them into the same cluster before considering them as co-referential. This establishes a higher threshold for merging similar mention words.

Therefore, if two mention words are highly ambiguous and have a lower similarity score (e.g. both have a score of around 0.45), they will be treated as separate co-reference clusters due to the high threshold (0.8-0.9). This ensures that our method can robustly handle ambiguous context issues. Furthermore, for ambiguous mentions, we treat them as an independent co-reference cluster. This means that such mentions will not be replaced in subsequent processing, and their interpretation will rely on the background knowledge encoded in the original model.

We provide a mechanism to simulate errors. Among all mentions, we randomly replace 5% of them, with a replacement range of ±10, to evaluate the impact of ambiguous references on the entire reasoning process. This allows us to assess how this situation affects the robustness and reliability of our method. Table 2 shows experimental results about perturbation. It can be observed that after applying the disturbance, performance decreases, and the results differ from the previous baseline. This suggests the importance of coreference resolution in the reasoning process, as model performance can be impacted by incorrect coreference resolution.

## E    FUTURE SCOPE

Our LQCA implementation handles long coreference relations based on language model inference in short texts. In long texts, a potential optimization direction is using causal inference methods. The causal relationship measure formula $C(e_i, e_j) = \alpha \cdot R_{context}(e_i, e_j) + \beta \cdot R_{causal}(e_i, e_j)$ combines traditional entity relationship measures with causal inference. The traditional entity relationship measure $R_{context}(e_i, e_j)$ is based on factors such as syntactic distance, entity type matching, and co-occurrence, while the causal relationship measure $R_{causal}(e_i, e_j)$ considers whether entities $e_i$ and $e_j$ are in a causal chain or connected through causal events. The weight coefficients $\alpha$ and $\beta$ control the influence of these two parts. The causal relationship measure $R_{causal}(e_i, e_j)$ reflects the causal dependencies between entities and can be calculated as follows:

First, based on a chain of events with causal connections, if the events $E_i$ and $E_j$ associated with entities $e_i$ and $e_j$ have a causal relationship, and if $E_j$ is the causal consequence of $E_i$, then $R_{causal}(e_i, e_j)$ will be large; for example, in the case of "The storm caused the flood," there is a strong causal relationship between the storm and the flood. Next, using causal inference models, such as Bayesian networks or causal graphs, the causal relationships between entities can be inferred from the model structure; for example, in a graph neural network, the relationships between entities can be deduced from the network structure and used to calculate $R_{causal}(e_i, e_j)$. Causal inference based on causal lexicons: causal connecting words in the text help to mark the causal relationships between events. When one entity triggers another, the causal relationship measure will be high.

To further enhance the reasoning of relationships between entities, causal relationship constraints are added to the entity relationship network, ensuring that connections are based not only on similarity but also on causal inference. This helps improve the accuracy of reasoning. Some entities may be at the cause or effect positions in a causal chain. Changes in entities can also be triggered by events,

such as a disease triggering a change in health status. This causal relationship can be encoded through a causal model and assist in reasoning between entities. We expect that better coreference could assist the model to understand the context better and provide a robost result on various cases.

## F  ETHICS STATEMENTS

Our work is based on open-source datasets and code for experimentation. All data and information comply with relevant code standards and data regulations, ensuring that there is no risk of privacy breaches or information leaks. The use of large language models in our paper is primarily applied to handling and solving long context understanding problem in most scenarios, as well as text processing during Chinese-to-English translation in token level. This fully complies with the conference's requirements and privacy security guidelines.

When interacting with large language models, we may utilize relevant information from instruction. It is important to note that hallucinations from large language models may lead to incorrect answers. Our approach can be further integrated into other frameworks.

## G  LONGEST DOT PRODUCT PATH ALGORITHM

We use this algorithm to compute the distance of mentions that calculates the distance across the entire graph. The corresponding algorithm updates paths in a manner similar to Dijkstra's algorithm. By extracting the node with the smallest distance from the priority queue, the algorithm traverses its neighboring nodes, with the selection of these neighbors constrained by a range parameter to ensure that only nodes within a given distance are considered. For each neighbor, the algorithm determines whether to update the optimal path from the current node to that neighbor by calculating the possible path lengths. If the newly calculated path length is greater than the currently recorded optimal path, an update is made accordingly.

---

**Algorithm 1** All-Pairs Longest Path for Restricted Graph

---

**Input:** Graph: $G = (V, E)$; Weight function: $w : E \rightarrow \mathbb{R}^+$; Range parameter: $k$
**Output:** Longest dot product path distances: $d_{i,j}$ for all $i, j \in V$

1: **for** each node $n \in V$ **do**             ▷ // Initialize distances for node $n$
2:  **for** each node $m \in V$ **do**
3:   **if** $m = n$ **then**
4:    $d_{n,m} \leftarrow 1$            ▷ // Distance to itself is 1
5:   **else**
6:    $d_{n,m} \leftarrow 0$           ▷ // Initial distance set to 0
7:   **end if**
8:  **end for**
9:  $Q \leftarrow V$         ▷ // Priority queue to store unvisited nodes
10:  **while** $Q$ is not empty **do**         ▷ // Dijkstra's algorithm
11:   $u \leftarrow$ Extract-Max$(Q)$      ▷ // Node with the smallest distance
12:   **for** $v \in \{m \in V \mid |m - u| \leq L, (u, m) \in E\}$ **do**  ▷ // Neighbors within range
13:    $alt \leftarrow d_{n,u} \times w(u,v)$     ▷ // Calculate potential distance
14:    **if** $alt > d_{n,v}$ **then**
15:     $d_{n,v} \leftarrow alt$       ▷ // Update distance if longer
16:    **end if**
17:   **end for**
18:  **end while**
19: **end for**
20: **return** $d_{i,j}$ for all $i, j \in V$    ▷ // Return all-pairs longest path distances

---

