# OpenReview forum: "Bridging Context Gaps: Leveraging Coreference Resolution for Long Contextual Understanding"
_ICLR.cc/2025/Conference — ICLR 2025 Poster_

### Official Review · Reviewer_CakD · 2024-10-31

**Soundness:** 3
**Presentation:** 2
**Contribution:** 3
**Rating:** 6
**Confidence:** 3

**Summary:**

This paper introduces the LQCA method, which aims to enhance the performance of LLMs in understanding lengthy contexts and executing effective question answering. The LQCA method focuses on coreference resolution within long contexts, systematically processing information to replace ambiguous or complex references with clearer mentions, thereby improving the model's comprehension. The paper claims that this approach leads to significant improvements in question answering performance on various LLMs and datasets, particularly on models like OpenAI-o1mini and GPT-4o.

**Strengths:**

The paper addresses an important issue in natural language processing, which is the difficulty LLMs face in dealing with long contexts.

The proposed LQCA method is a novel approach that combines coreference resolution with question answering in long contexts.

**Weaknesses:**

While the paper demonstrates improvements in performance, it does not provide a comprehensive comparison with alternative methods that also aim to improve long-context understanding.

The paper could benefit from a more detailed discussion on how the LQCA method handles cases with highly ambiguous references, which could be a common occurrence in real-world applications.

There is a lack of analysis on the computational overhead introduced by the LQCA method, which is crucial for understanding its scalability and practical applicability.

**Questions:**

How does the LQCA method perform compared to other state-of-the-art approaches in terms of accuracy and efficiency, especially when dealing with extremely long or complex texts?

How does the LQCA method handle highly ambiguous contexts where multiple interpretations of coreferences are possible?

What are the potential impacts of incorrect coreference resolution on the quality of question answering, and how does the LQCA method address these issues?

Can the authors provide more insights into the computational overhead introduced by the LQCA method and its scalability for even longer contexts?

---

> ### Author Response · Authors · 2024-11-25
> **Response to Reviewer CakD(1/2)**
>
> Thank you to the reviewers for their evaluation of our paper and for offering valuable suggestions. I am particularly apprecitated for the praise, such as recognizing our discussion of a significant issue and highlighting that our proposed method is a novel approach. Next, I will provide detailed responses to the questions raised by you.
>
> > __W1&Q1__: While the paper demonstrates improvements in performance, it does not provide a comprehensive comparison with alternative methods that also aim to improve long-context understanding. How does the LQCA method perform compared to other state-of-the-art approaches in terms of accuracy and efficiency, especially when dealing with extremely long or complex texts?
>
> In fact, our method compares various aspects of context understanding and long-context processing explored in current research. Existing methods for handling long contexts can generally be categorized into two main types.
>
> The first category involves architectural improvements to models, such as updating attention mechanisms or compressing data or textual information through encoding. These approaches require integration into the model itself. However, given the current scenario where many large language models (LLMs) are closed-source, modifying attention mechanisms or model architectures may not be well-suited for end-to-end scenarios of this type.
>
> The second category includes retrieval-augmented or tool-assisted learning methods. These methods aim to enable the model to identify and extract the most relevant content within a passage, often with the help of external tools, and then enhance the LLM's focus on this extracted content. In retrieval-augmented contexts, the retrieved text segments often contain partial information or answers directly relevant to downstream tasks. By condensing the extensive redundant information in long contexts into concise, answer-focused segments, these methods help improve the model’s comprehension of long-context passages.
>
> Based on our experiments, we conducted comparisons between our approach and various cutting-edge methods for handling long-text contexts, including prompting, generation, retrieval, and augmentation-based methods. The results across multiple metrics demonstrate significant improvements with our method and highlight an important issue: appropriate coreference resolution is crucial for LLMs in understanding, generating, and tackling downstream tasks. Since LLMs often exhibit deficiencies in coreference resolution, enhancing this capability can substantially improve their performance in various question-answering scenarios and other downstream tasks.
>
> > __W2&Q2&Q3__. The paper could benefit from a more detailed discussion on how the LQCA method handles cases with highly ambiguous references, which could be a common occurrence in real-world applications. How does the LQCA method handle highly ambiguous contexts where multiple interpretations of coreferences are possible? What are the potential impacts of incorrect coreference resolution on the quality of question answering, and how does the LQCA method address these issues?
>
> First, our method demonstrates an effective filtering mechanism for handling highly ambiguous references. Specifically, in our algorithm, when the similarity between two mentions exceeds a set threshold (e.g., 0.9), at least 9 out of 10 sub-documents containing both mentions must classify them into the same cluster before they are considered coreferent. This establishes a relatively high threshold for merging similar mentions.
>
> As a result, if two mentions are highly ambiguous and their similarity is relatively low (e.g., both scoring around 0.45), they would remain as separate coreference clusters due to the high threshold of 0.8–0.9. This ensures that our method robustly handles ambiguous contextual issues. Furthermore, in cases where a single mention appears ambiguously, it is treated as its own coreference cluster. This means that such mentions are not replaced in later stages, and their interpretation relies on the background knowledge encoded in the original model.
>
> Nevertheless, we also provide a mechanism to simulate errors. For all mentions, we randomly replace their corresponding mentions within a ±10 range in 5% of cases to evaluate the impact of ambiguous references on the overall reasoning process. This allows us to assess how such cases affect the robustness and reliability of our method. The corresponding experimental results are shown in next windows below.

---

> ### Author Response · Authors · 2024-11-25
> **Response to Reviewer CakD(2/2)**
>
> > __W2&Q2&Q3_continue__:
>
> The experiment on mentions replacement:
>
> | Model          | Methods                           | LooGLE-arxiv-paper-abstract | L-Eval-TOEFL | L-Eval-Coursera | LongBench-HotpotQA | LongBench-2WikiMHQA |
> |---|-----|------|----|----|------|--|
> | OpenAI-o1-mini | RAG with self-Corpus              | 30.95                       | 86.55        | 83.64           | 70.26              | 48.23               |
> |                | LQCA-with 5% mentions replacement | 32.51                       | 88.22        | 82.98           | 71.66              | 47.55               |
> |                | LQCA     | 40.65                       | 93.71        | 87.25           | 75.92              | 56.85               |
> | GPT-4o         | RAG with self-Corpus              | 27.65                       | 85.13        | 80.65           | 66.35              | 47.01               |
> |                | LQCA-with 5% mentions replacement | 28.54                       | 82.51        | 81.35           | 68.77              | 47.23               |
> |     | LQCA                              | 34.53                       | 91.03        | 85.12           | 71.66              | 53.62               |
> | Llama-3-8b     | RAG with self-Corpus              | 6.84                        | 63.15        | 48.77           | 40.72              | 31.78               |
> |                | LQCA-with 5% mentions replacement | 6.56                        | 64.25        | 47.29           | 41.29              | 33.69               |
> |                | LQCA                              | 6.91                        | 69.55        | 49.15           | 46.23              | 34.72               |
>
> It can be observed that after applying perturbations, the performance declines, with results varying compared to the previous baseline. This indicates that coreference resolution is relatively important, as the model's performance may degrade due to incorrect coreference resolution (which is also an issue when raw text is used without resolution).
>
> > __Q4__: There is a lack of analysis on the computational overhead introduced by the LQCA method, which is crucial for understanding its scalability and practical applicability. Can the authors provide more insights into the computational overhead introduced by the LQCA method and its scalability for even longer contexts?
>
> The computational overhead of our method mainly involves the complexity of two processes. The first is the complexity of coreference resolution for sub-documents, handled during the Maverick process. The second is the complexity of distance computation.
>
> For the distance computation, the complexity is primarily influenced by the sliding window length. In merging long-distance coreferences, the complexity is $O(nlogn)$, where $n$ represents the number of mentions. During the process of finding strongly connected components, the overall resolution can be achieved using Tarjan's strongly connected component algorithm, which has a complexity of $O(n)$ due to its depth-first search approach. Thus, the overall complexity remains within a manageable range.
>
> Empirically, we compared the average latency(in 8 nvidia-A100, with batch 64) between directly querying the large language model and using our LQCA method for reasoning. The corresponding results are shown in the table below.
>
> | Models | Methods    | LooGLE-arxiv-paper-abstract | L-Eval-TOEFL | L-Eval-Coursera | LongBench-HotpotQA | LongBench-2WikiMultihopQA |
> |--------|------------|-----------------------------|--------------|-----------------|--------------------|---------------------------|
> | gpt-4o | Vanilla LM | 12.83s                      | 5.96s        | 8.23s           | 9.19s              | 5.82s                     |
> |        | LQCA       | 16.21s                      | 7.28s        | 10.56s          | 11.81s             | 7.55s                     |
>
>
> From the results, we can see that our method achieves excellent performance across various downstream and long-text tasks with limited additional latency. This further demonstrates that replacing mentions in long-text scenarios with their more accurate coreferent substitutes is highly beneficial. This is particularly true for pronouns such as "he," "she," or "it," which lack explicit references; our method enables more precise identification of the intended target and retrieval of the correct answer.
>
> Since the complexity of our method is primarily related to the context length and the distribution of mentions, it can adapt well to scenarios with longer contexts without significantly increasing computational complexity.
>
> Thank you once again for your feedback on our paper. This will definitely help us improve our work. If these suggestions have helped clarify any doubts you had, we would appreciate it if you could improve the ratings. If there are any unclear points or further details you'd like to discuss, please feel free to respond.

---

> > ### Comment · Reviewer_CakD · 2024-11-26
> >
> > Thank you for your feedback. That addresses my problem. I will raise my score to 6.

---

### Official Review · Reviewer_syvQ · 2024-11-03

**Soundness:** 3
**Presentation:** 3
**Contribution:** 2
**Rating:** 6
**Confidence:** 4

**Summary:**

This paper investigates LLM performance in question answering and summarization tasks with very long
inputs (128K). The hypothesis put forward is that preprocessing the input with coreference
resolution allows the model to understand the long context better, in particular to identify
and manage references effectively. The proposed method  encompasses
four key steps: resolving coreferences within sub-documents, computing the distances between mentions,
 defining a representative mention for coreference, and
answering questions through mention replacement. Experimental evaluations on a range of LLMs and
datasets yield overall improvement.

**Strengths:**

- The idea of performing coreference resolution before attempting to do the task at hand, makes sense and is corroborated
by empirical results showing improvements across models and datasets.

-  The paper is well-written and the proposed method easy to follow.

- The proposed coreference approach seems intuitive and could be also used during LLM pertaining or fine-tuning.

**Weaknesses:**

- The technical contribution is somewhat limited. Coreference merging is interesting, however, without empirical evaluation it is not possible to tell whether it actually works (only down-stream evaluation results are presented, see my questions below).

- What is the main contribution of this paper, is it the algorithm in Section 3.2?

-

**Questions:**

- Please explain what I am supposed to be seeing in Figure 2. Your caption is not very informative either.

- It would been useful to see  an actual example for the method in Section 3.2.

- Please present the prompts you use for the various tasks (e.g., QA, summarization), in the appendix.

- What is the accuracy of your coreference algorithm with the mention merging computation? How accurate is your approach?
 I understand that the input is very long, but perhaps you could perform some manual evaluation on shorter documents. It would be interesting to see what the degree of noise is, and how much better your results would be if coreference was perfect.

- Please describe Maverick in more detail, ILCR readers may not be familiar with what it does.

- I would have liked to see if results improve further with fine-tuning (e.g., using Lora).

- What is the computational complexity of the coreference preprocessing step?
- RAG and RECOMP are not incompatible with your method, could you not run coreference on the retrieved segments?

- It would be good to analyze why you observe performance improvements. Is it because you identify named entities more accurately?

- Existing work (e.g., https://aclanthology.org/2024.lrec-main.145.pdf) shows that LLMs are not very good at coreference resolution, however, it would have been more reasonable to give the LLM some examples (my understanding is you try to perform coreference in a zero-shot setting).

-  Is your Avg column in table 1 averaging across Rouge, Accuracy, and F1?

-

---

> ### Author Response · Authors · 2024-11-25
> **Response to Reviewer syvQ(1/4)**
>
> Thank you very much to the reviewer for evaluating our paper and for praising certain aspects of it. This includes recognizing that the ideas presented are meaningful and supported by empirical results. The paper is well-written, the proposed method is easy to understand, and it is intuitive and applicable across various scenarios. I will now provide detailed explanations in response to the issues you have raised.
>
> > __W1__: The technical contribution is somewhat limited. Coreference merging is interesting, however, without empirical evaluation it is not possible to tell whether it actually works (only down-stream evaluation results are presented, see my questions below).
>
> In fact, the technical contribution of our paper lies in providing an algorithmic approach to merging coreference resolution for long texts. In previous scenarios, coreference resolution tasks were often limited to fine-tuned encoder models like the BERT series, which inherently restricted the contextual length. Moreover, directly performing coreference resolution on extended contexts significantly underperforms these specialized methods. As you mentioned, current studies reveal that large language models struggle with coreference resolution tasks. Thus, addressing coreference resolution in long texts and investigating whether it can enhance reasoning in large language models becomes critically important.
>
> Furthermore, a series of prior works have shown that large language models are highly sensitive to specific coreference and entity-related information, particularly in tasks involving entity and relationship-based intent recognition. This sensitivity greatly impacts task-solving performance. From these perspectives, our work provides new support for processing and analyzing long texts, offering a transferable framework for coreference resolution. When the coreference resolution capability of smaller models improves, our framework allows this capability to be transferred to long-text contexts, thereby improving performance in both coreference resolution and downstream tasks.
>
> > __W2__: What is the main contribution of this paper, is it the algorithm in Section 3.2?
>
> This paper makes two primary contributions. The first is the handling of coreference resolution in long texts, detailed in Sections 3.1 to 3.3. This involves merging mentions across document slices derived from long texts. Such mention merging poses significant challenges in previous methods due to inconsistencies in coreference resolution across different sub-documents. Our approach, however, integrates information from various sub-documents to select the optimal coreference strategy. For ambiguous cases, we retain the original text, which substantially improves the accuracy of coreference resolution in extended contexts.
>
> Second, we validate the transferability of the coreference resolution results generated by the Maverick algorithm to long texts. This validation is conducted through downstream tasks, where we assess the model's ability to leverage these results. The experiments demonstrate that incorporating more specific contextual information improves performance across various downstream tasks. This finding suggests that the capabilities of dialogue systems in handling coreference information can be effectively transferred to large language models, enabling them to better understand mentions and their relationships, thereby enhancing their ability to solve downstream tasks.
>
> Moreover, our work represents the first study focusing on coreference resolution in long texts and the first to evaluate downstream task performance after applying coreference resolution within large language models. This highlights the critical role of coreference resolution in improving the reasoning capabilities of large language models.
>
> > __Q1__: Please explain what I am supposed to be seeing in Figure 2. Your caption is not very informative either.
>
> The middle part of Figure 2 illustrates the three steps we employ in the coreference resolution process, corresponding to Sections 3.1 to 3.3. This process involves slicing the context into fragments, applying the Maverick algorithm to each fragment to extract mentions and coreferences. Based on the mentions and coreferences in these sub-documents, we calculate intra-document and inter-document distances. When the distance exceeds a predefined threshold, we consider the two mentions to belong to the same coreference cluster. Finally, we identify the optimal coreference cluster, which replaces the original mentions in the context.
>
> The overall workflow in Figure 2 represents this process for long texts. The bottom of the figure depicts how the modified text, with resolved coreferences, is used as input for downstream tasks to perform reasoning. Key aspects to focus on include the probability calculation between different mentions within the document and the distance-based slicing of sub-documents.

---

> ### Author Response · Authors · 2024-11-25
> **Response to Reviewer syvQ(2/4)**
>
> > __Q2__: It would been useful to see an actual example for the method in Section 3.2.
>
> Yes, I completely agree with this point. The core issue lies in determining whether a specific mention in a subdocument belongs to the same coreference cluster as other mentions within the same subdocument. This process is decided by calculating the overall probability of coreference between two mentions to determine their coreference probability within the same subdocument. As for cross-document mention relationships, this is primarily computed through an algorithmic process. We will provide a detailed explanation of the specific algorithm for individual mention calculations in the appendix.
>
> > __Q3__: Please present the prompts you use for the various tasks (e.g., QA, summarization), in the appendix.
>
> We would include them in the appendix, but in fact, our approach does not require extensive prompt information. For the LQCA framework, the main focus is on replacing the original text. Once the original text is replaced, the reasoning, downstream task prompts, and methods used by the vanilla LM remain the same.
>
> > __Q4__: What is the accuracy of your coreference algorithm with the mention merging computation? How accurate is your approach? I understand that the input is very long, but perhaps you could perform some manual evaluation on shorter documents. It would be interesting to see what the degree of noise is, and how much better your results would be if coreference was perfect.
>
>
> Our coreference algorithm and the accuracy of volume-based merging calculations are comparable to Maverick's algorithm in the final computation. Considering that there are currently no established metrics for evaluating tape-based resolution algorithms for long texts, we addressed your requirements by manually annotating ten samples each from small subsets of three large datasets. However, we acknowledge that manual annotation may introduce some distortion.
>
> To mitigate this, we also evaluated coreference resolution performance on the OntoNotes benchmark, which is widely used for short-document coreference resolution. Furthermore, we conducted experiments not only on long documents but also by concatenating questions from OntoNotes into documents with lengths of approximately 10,000 tokens to assess whether the overall coreference resolution capability exhibited any distortions. The results are as follows.
>
> |                           | OntoNotes | Loogle-Arxiv paper abstract | LooGLE dependency QA | L-Eval TOEFL | L-Eval Coursera | LongBench HotpotQA | LongBench 2WikiMultihopQA |
> |---------------------------|-----------|-----------------------------|----------------------|--------------|-----------------|--------------------|---------------------------|
> | LQCA Coreference Accuracy | 80.6      | 92.2                        | 89.5                 | 91.3         | 93.5            | 93.2               | 91.7                      |
>
> Experimental results demonstrate that our method performs well in most scenarios. The slight decrease in accuracy for certain cases is primarily due to some mentions being overly ambiguous. In such instances, retaining the original text for subsequent reasoning proves to be a more reliable approach to ensure overall accuracy.
>
> We also evaluated the performance of mentions aligned with manual annotations during inference,
> To primarily evaluate the quality of text generation, we conducted comparative experiments on the examples we annotated using LooGLE and compared their corresponding F1 scores. and the experimental results are as follows:
>
> | Model          | Methods                        | LooGLE-arxiv-paper-abstract | LooGLE dependency QA |
> |----------------|--------------------------------|-----------------------------|----------------------|
> | OpenAI-o1-mini | RAG with self-Corpus           | 28.53                       | 30.23                |
> |                | LQCA                           | 38.99                       | 36.88                |
> |                | Manual-coreference replacement | **41.95**                   | **38.57**            |
> | GPT-4o         | RAG with self-Corpus           | 28.33                       | 29.45                |
> |                | LQCA                           | 32.12                       | **31.56**            |
> |                | Manual-coreference replacement | **35.87**                   | 30.79                |
> | Llama-3-8b     | RAG with self-Corpus           | 7.89                        | 7.33                 |
> |                | LQCA                           | 8.55                        | 9.03                 |
> |                | Manual-coreference replacement | **10.32**                   | **9.68**             |
>
> when various coreferences align with human logic, the metrics show noticeable improvement. This indicates that large language models maintain strong problem-solving capabilities when dealing with high-quality text.

---

> ### Author Response · Authors · 2024-11-25
> **Response to Reviewer syvQ(3/4)**
>
> > __Q5__: Please describe Maverick in more detail, ICLR readers may not be familiar with what it does.
>
> Maverick is a fine-tuned pre-trained model based on DeBERTa, specifically designed for coreference resolution. It consists of four main components: mention extraction, mention regularization, mention pruning, and clustering. By processing documents and training a global loss function for clustering, it generates the final results for coreference resolution.
>
> Maverick provides the following key information: the positions of mentions within the document and the cluster to which each mention belongs. These details encompass the essential information required for performing coreference resolution in documents of standard length.
>
> > __Q6__: I would have liked to see if results improve further with fine-tuning (e.g., using Lora).
>
> Our approach extends coreference information from short documents to long documents. For LoRA-based fine-tuning, the process involves replacing all coreference-related information in the large language model through input-output training data. This requires both the original unmodified text segments as input and the modified text segments as output.
>
> To ensure overall training quality, we used OntoNotes data as ground truth for fine-tuning. The coreference accuracy results on the annotated long-text datasets after fine-tuning are as follows:
>
> |                           | Loogle-Arxiv paper abstract | LooGLE dependency QA | L-Eval TOEFL | L-Eval Coursera | LongBench HotpotQA | LongBench 2WikiMultihopQA |
> |---------------------------|-----------------------------|----------------------|--------------|-----------------|--------------------|---------------------------|
> | LQCA Coreference Accuracy | 92.2                        | 89.5                 | 91.3         | 93.5            | 93.2               | 91.7                      |
> | LoRA on Llama-3-8b        | 78.6                        | 77.3                 | 82.5         | 81.1            | 76.3               | 80.5                      |
>
> it is shown that the fine-tuning results are generally inferior to our method. This is primarily due to the inherent complexity of the task, as identifying all mentions and replacing their references step by step is overly intricate. Fine-tuning may not handle such tasks flawlessly.
>
> Therefore, our method represents a stable and robust approach by leveraging the Maverick algorithm to obtain coreference results for short documents and extending them to long documents effectively.
>
> > __Q7__: What is the computational complexity of the coreference preprocessing step?
>
> The computational overhead of our method mainly involves the complexity of two processes. The first is the complexity of coreference resolution for sub-documents, handled during the Maverick process. The second is the complexity of distance computation.
>
> For the distance computation, the complexity is primarily influenced by the sliding window length. In merging long-distance coreferences, the complexity is $O(nlogn)$, where $n$ represents the number of mentions. During the process of finding strongly connected components, the overall resolution can be achieved using Tarjan's strongly connected component algorithm, which has a complexity of $O(n)$ due to its depth-first search approach. Thus, the overall complexity remains within a manageable range.
>
> Empirically, we compared the average latency(in 8 nvidia-A100, with batch 64) between directly querying the large language model and using our LQCA method for reasoning. The corresponding results are shown in the table below.
>
> | Models | Methods    | LooGLE-arxiv-paper-abstract | L-Eval-TOEFL | L-Eval-Coursera | LongBench-HotpotQA | LongBench-2WikiMultihopQA |
> |--------|------------|---------|--------------|-----------------|--------|-------------|
> | gpt-4o | Vanilla LM | 12.83s                      | 5.96s        | 8.23s           | 9.19s              | 5.82s                     |
> |        | LQCA       | 16.21s                      | 7.28s        | 10.56s          | 11.81s             | 7.55s                     |
> |        | preprocessing       | 3.38s                      | 1.32s        | 2.23s          | 2.62s             | 1.73s                     |
>
> From the results, we can see that our method achieves excellent performance across various downstream and long-text tasks with limited additional latency. This further demonstrates that replacing mentions in long-text scenarios with their more accurate coreferent substitutes is highly beneficial. This is particularly true for pronouns such as "he," "she," or "it," which lack explicit references; our method enables more precise identification of the intended target and retrieval of the correct answer.
>
> Since the complexity of our method is primarily related to the context length and the distribution of mentions, it can adapt well to scenarios with longer contexts without significantly increasing computational complexity.

---

> > ### Comment · Reviewer_syvQ · 2024-11-26
> >
> > Thank you for addressing my concerns, and for the extra experiments! I will raise my score.

---

> ### Author Response · Authors · 2024-11-25
> **Response to Reviewer syvQ(4/4)**
>
> > __Q8__: RAG and RECOMP are not incompatible with your method, could you not run coreference on the retrieved segments?
>
> First, we introduced RAG and similar components to compare our coreference resolution method with commonly used retrieval-augmented methods in context learning for long-text tasks such as summarization and QA. As shown in Table 1, our LQCA method outperforms the naive approach of slicing and retrieving document segments, achieving significant performance improvements.
>
> In the ablation study, we aimed to determine whether providing only a single coreference-resolved document slice, instead of the entire document, would suffice. The ablation results indicate that providing the entire document's information is more beneficial for downstream tasks than offering just a single slice.
>
> Through this process, we mainly sought to compare and highlight the importance of text quality and coreference resolution outcomes in long-text QA. These aspects are crucial as they provide the model with deterministic information, enabling it to better locate and address the problem at hand.
>
> > __Q9__: It would be good to analyze why you observe performance improvements. Is it because you identify named entities more accurately?
>
> The superior performance of our method can be attributed to the enhanced specificity in coreference resolution within text segments. Based on the current experimental results, we believe that our approach enables a more precise integration of knowledge related to corresponding mentions, ultimately leading to better task outcomes for downstream applications.
>
> In contrast to naive retrieval-augmented methods and zero-shot chain-of-thought approaches, these alternatives only reduce the density of information to some extent but fail to improve the intrinsic quality of the text. Providing effective coreference resolution results facilitates a deeper understanding of contextual content, which is why our method excels at addressing problems in long-text scenarios with greater accuracy.
>
> > __Q10__: Existing work (e.g., https://aclanthology.org/2024.lrec-main.145.pdf) shows that LLMs are not very good at coreference resolution, however, it would have been more reasonable to give the LLM some examples (my understanding is you try to perform coreference in a zero-shot setting).
>
> Yes, large language models perform relatively poorly in coreference resolution, which is why our method requires an algorithmic approach to handle coreference resolution for long texts. Specifically, we extract coreference information from slices of the document as short texts and extend it to the entire long document. In the ablation study, we also experimented with directly performing coreference resolution on long texts. As shown in Figure 3, the results indicate that using an LLM to directly resolve coreferences in long texts and then perform downstream reasoning tasks yields outcomes that are significantly less effective compared to our framework. However, this approach does improve text quality to some extent when compared to directly providing the raw text to the large language model for task responses.
>
> > __Q11__: Is your Avg column in table 1 averaging across Rouge, Accuracy, and F1?
>
>
> Yes. The "average" column in Table 1 represents the average of these values. Our primary goal is to observe the overall average performance of the corresponding methods. Although the metrics differ, they still provide a useful point of comparison, demonstrating that our method consistently outperforms previous approaches and achieves the best results across multiple metrics and datasets.
>
> Thank you once again for your feedback on our paper. This will definitely help us improve our work. If these suggestions have helped clarify any doubts you had, we would appreciate it if you could improve the ratings. If there are any unclear points or further details you'd like to discuss, please feel free to respond.

---

### Official Review · Reviewer_ZAZy · 2024-11-04

**Soundness:** 3
**Presentation:** 3
**Contribution:** 3
**Rating:** 8
**Confidence:** 3

**Summary:**

This work proposes to disambiguate mentions in a document by replacing pronouns with corresponding disambiguated entities identified by coreference resolution so that a large language model (LLM) can easily leverage a long context document for better contextual understanding. One of the crucial problem for an LLM is to understand a long context to respond to a query, but ambiguities in mentions in a document can easily impact the understanding to derive an answer. This work employs a BERT-based model to identify coreference resolution as a preprocessing step to modify a document-wise context to disambiguate mentions. Experiments are carried out on diverse tasks demanding long context understanding, e.g., LongBench, and show large gains when compared with other methods. e.g., RAGs.

**Strengths:**

- A simple method to resolve ambiguity of mentions in a document so that an LLM can easily leverage the contextual information to generate a response. The use of a BERT-based model is a limitation in that it cannot handle longer contexts, but this work proposes a method to combine coreference resolution results for sub-documents using a simple statistics, i.e., whether a mention is within a cluster or not, to compute a distance metric.

- Experiments show gains when compared with other methods considering long contexts, e.g., RAGs. Analysis is also interesting enough to quantify the gains of the proposed method.

**Weaknesses:**

- It is not clear how the errors in coreference resolution might have an impact to the end performance since coreference resolution itself is a rather challenging task.

**Questions:**

I'd like to know how the errors in coreference resolution might have an impact to the end performance. Probably it is possible to inject noises in coreference clusters to see how that will impact to the end performance.

---

> ### Author Response · Authors · 2024-11-25
> **Response to Reviewer ZAZy**
>
> Thank you very much for reviewing our paper, and we especially appreciate your kind words about its content. As a simple approach to addressing the remaining issues in the document, our experiments demonstrate that the method offers distinct advantages, with analyses that are sufficiently engaging and capable of quantifying the benefits of the proposed approach. I will now provide a detailed explanation addressing the weaknesses you have identified.
>
> > __W1&Q1__: It is not clear how the errors in coreference resolution might have an impact to the end performance since coreference resolution itself is a rather challenging task. I'd like to know how the errors in coreference resolution might have an impact to the end performance. Probably it is possible to inject noises in coreference clusters to see how that will impact to the end performance.
>
>
> First, our method demonstrates an effective filtering mechanism for handling highly ambiguous references. Specifically, in our algorithm, when the similarity between two mentions exceeds a set threshold (e.g., 0.9), at least 9 out of 10 sub-documents containing both mentions must classify them into the same cluster before they are considered coreferent. This establishes a relatively high threshold for merging similar mentions.
>
> As a result, if two mentions are highly ambiguous and their similarity is relatively low (e.g., both scoring around 0.45), they would remain as separate coreference clusters due to the high threshold of 0.8–0.9. This ensures that our method robustly handles ambiguous contextual issues. Furthermore, in cases where a single mention appears ambiguously, it is treated as its own coreference cluster. This means that such mentions are not replaced in later stages, and their interpretation relies on the background knowledge encoded in the original model.
>
> Nevertheless, we also provide a mechanism to simulate errors. For all mentions, we randomly replace their corresponding mentions within a ±10 range in 5% of cases to evaluate the impact of ambiguous references on the overall reasoning process. This allows us to assess how such cases affect the robustness and reliability of our method. The corresponding experimental results are shown below.
>
> The experiment on mentions replacement:
>
> | Model          | Methods                           | LooGLE-arxiv-paper-abstract | L-Eval-TOEFL | L-Eval-Coursera | LongBench-HotpotQA | LongBench-2WikiMHQA |
> |---|-----|------|----|----|------|--|
> | OpenAI-o1-mini | RAG with self-Corpus              | 30.95                       | 86.55        | 83.64           | 70.26              | 48.23               |
> |                | LQCA-with 5% mentions replacement | 32.51                       | 88.22        | 82.98           | 71.66              | 47.55               |
> |                | LQCA     | 40.65                       | 93.71        | 87.25           | 75.92              | 56.85               |
> | GPT-4o         | RAG with self-Corpus              | 27.65                       | 85.13        | 80.65           | 66.35              | 47.01               |
> |                | LQCA-with 5% mentions replacement | 28.54                       | 82.51        | 81.35           | 68.77              | 47.23               |
> |     | LQCA                              | 34.53                       | 91.03        | 85.12           | 71.66              | 53.62               |
> | Llama-3-8b     | RAG with self-Corpus              | 6.84                        | 63.15        | 48.77           | 40.72              | 31.78               |
> |                | LQCA-with 5% mentions replacement | 6.56                        | 64.25        | 47.29           | 41.29              | 33.69               |
> |                | LQCA                              | 6.91                        | 69.55        | 49.15           | 46.23              | 34.72               |
>
> It can be observed that after applying perturbations, the performance declines, with results varying compared to the previous baseline. This indicates that coreference resolution is relatively important, as the model's performance may degrade due to incorrect coreference resolution (which is also an issue when raw text is used without resolution).
>
> Thank you once again for your feedback on our paper. This will definitely help us improve our work. If these suggestions have helped clarify any doubts you had, we would appreciate it if you could improve the ratings. If there are any unclear points or further details you'd like to discuss, please feel free to respond.

---

> > ### Comment · Reviewer_ZAZy · 2024-11-26
> >
> > Thank you very much for providing additional explanation and experiments. Your further explanation regarding the algorithm help me understand the motivation much clearer and definitely it should be included in the revised version. I think the extra experiment further strengthen the proposed method.

---

### Author Response · Authors · 2024-12-03
**General Response for Readers**

We sincerely appreciate the thorough review and valuable suggestions provided by everyone on our paper. We are also deeply grateful to the reviewers for their insightful comments and recognition of our work. We believe these suggestions not only contribute to enhancing the quality of our paper but also have a profound impact on advancing the research on the proposed method. Here, we wish to reiterate the strengths and uniqueness of our study to assist readers in gaining a more comprehensive understanding of our approach while addressing common concerns more explicitly.

> 1: A New Perspective and Approach to Long Contextual Understanding

Our method offers a novel solution to long-text understanding. Previous approaches often relied on improving the base models or leveraging retrieval-augmented and tool-integrated methods to aid context understanding and access relevant information. While effective, these methods frequently overlook the quality of the text itself. In traditional scenarios, such as dialogue systems, many information extraction techniques effectively capture the core content of an article.

Our research demonstrates that coreference resolution not only enhances the performance of large language models (LLMs) in processing long texts and handling coreference but also improves the overall quality of the text. This highlights the auxiliary potential of information extraction techniques in future LLMs.

> 2: An Innovative Coreference Resolution Approach for Long Texts

Traditional coreference resolution methods are predominantly designed for short texts, mainly due to the input length limitations of models like BERT. Consequently, many coreference resolution models can only function within the confines of short text ranges. Our research overcomes this limitation. Our method integrates coreference information across different segments of a long document and consolidates mentions belonging to the same coreference set. Even for mentions that are far apart in a long text, we propose a merging scheme based on distance information.

This innovation has demonstrated significant improvements across various downstream tasks, including long-text summarization and question answering. Our study marks an important advancement in the field of coreference resolution for long texts.

> 3: Comprehensive Experimental Analysis and Performance on SOTA LLMs

Through detailed ablation studies and fine-grained analysis of problem placements, we have showcased the potential of our method. Experimental results indicate that our coreference resolution approach effectively mitigates the "contextual confusion" issues that LLMs often encounter when processing medium-to-long texts. Compared to direct retrieval-augmented segmentation methods, our approach is not only more efficient and convenient but also significantly enhances task performance.

__Common Problems from Reviewers:__

> 1: how the errors in coreference resolution might have an impact to the end performance.

Our method demonstrates an effective filtering mechanism for handling highly ambiguous references. if two mentions are highly ambiguous and their similarity is relatively low (e.g., both scoring around 0.45), they would remain as separate coreference clusters due to the high threshold of 0.8–0.9. This ensures that our method robustly handles ambiguous contextual issues. Furthermore, in cases where a single mention appears ambiguously, it is treated as its own coreference cluster. This means that such mentions are not replaced in later stages, and their interpretation relies on the background knowledge encoded in the original model.

After applying perturbations, the performance declines, with results varying compared to the previous baseline. This indicates that coreference resolution is relatively important, as the model's performance may degrade due to incorrect coreference resolution

> 2: What is the computational complexity?

The computational overhead of our method mainly involves the complexity of two processes. The first is the complexity of coreference resolution for sub-documents, handled during the Maverick process. The second is the complexity of distance computation.

For the distance computation, the complexity is primarily influenced by the sliding window length. In merging long-distance coreferences, the complexity is $O(nlogn)$, where $n$ represents the number of mentions. During the process of finding strongly connected components, the overall resolution can be achieved using Tarjan's strongly connected component algorithm, which has a complexity of $O(n)$ due to its depth-first search approach. Thus, the overall complexity remains within a manageable range.

We have provided detailed responses to more specific questions and experiments in our replies to the reviewers. We sincerely appreciate everyone's feedback on our paper and for contributing to the improvement of our work.

---

### Meta-Review · Area_Chair_fzy3 · 2024-12-21

**Metareview:**

(a) Scientific Claims and Findings
The paper claims that leveraging coreference resolution improves LLM performance in long-context tasks by reducing ambiguity and enhancing comprehension. The LQCA method systematically resolves and integrates coreferences, significantly boosting performance on downstream tasks like question answering.

(b) Strengths

- Novel approach for applying coreference resolution to long texts.
- Demonstrated improvements across various datasets and tasks.
- Detailed analysis and ablation studies highlight the robustness and scalability of the method.

(c) Weaknesses

- Limited discussion of computational overhead and efficiency in real-world scenarios.
- Lack of direct comparison with some alternative long-context understanding methods.
- Coreference accuracy and its standalone effectiveness are only indirectly evaluated.

(d) Reasons for Decision
The paper has strong empirical results and proposes a novel approach. The method’s clear potential and the significant improvements on downstream tasks warrant its acceptance.

**Additional Comments On Reviewer Discussion:**

During the rebuttal period, reviewers raised concerns about the computational complexity, the handling of ambiguous coreferences, and the standalone evaluation of coreference accuracy. The authors provided detailed responses, including additional experiments demonstrating the robustness of their method and its computational feasibility. The reviewers acknowledged the clarification and experiments, leading to improved scores. Reviewer syvQ specifically raised concerns about technical contribution but appreciated the responses and increased their score.

---

### Decision · Program_Chairs · 2025-01-22

Accept (Poster)